

# The impacts of assimilating Aeolus horizontal line-of-sight winds on numerical predictions of Hurricane Ida (2021) and a mesoscale convective system over the Atlantic Ocean

Chengfeng Feng[1], Zhaoxia Pu[1]

[1]Department of Atmospheric Sciences, University of Utah, Salt Lake City, 84112, United States

*Correspondence to*: Zhaoxia Pu (Zhaoxia.Pu@utah.edu)

**Abstract.** On 22 August 2018, the European Space Agency (ESA) launched the first spaceborne wind lidar, the Aeolus satellite, measuring horizontal line-of-sight (HLOS) winds globally. The assimilation of Aeolus HLOS winds has been proven to improve numerical weather predictions (NWPs). Still, its influences on forecasts of tropical cyclones (TCs) and

tropical convective systems have yet to be examined in detail. This study investigates the impacts of assimilating Aeolus HLOS winds on the analysis and forecasts of Hurricane Ida (2021) and a mesoscale convective system (MCS) embedded in an African easterly wave (AEW) over the Atlantic Ocean (AO) with the mesoscale community Weather Research and Forecasting (WRF) model and the NCEP-GSI based three-dimensional ensemble-variational (3DEnVAR) hybrid data assimilation (DA) system. Mie-cloudy and Rayleigh-clear winds are assimilated. The results for Ida (2021) show that

assimilating Aeolus HLOS winds leads to better track predictions. The intensity forecasts are improved in some cases, even with limited coverage of Aeolus HLOS winds within the inner core region of Ida (2021). In addition, the structure of heavy precipitation associated with Ida (2021) is refined after assimilation of Aeolus HLOS winds. Further diagnosis demonstrates that the improved intensity and precipitation forecasts result from enhanced divergence in the upper level of the troposphere after assimilation of Aeolus HLOS winds. Additional results from the MCS associated with an AEW indicate that

assimilating Aeolus HLOS winds enhances forecasts of its precipitation structure and the associated low-level divergence. In short, this study demonstrates the potential of assimilation of Aeolus HLOS winds to improve forecasts for TCs and tropical convective systems.

## 1 Introduction

Measuring three-dimensional wind profiles in the Global Observing System (GOS), especially over the oceans and remote

land areas, is essential for exploring atmospheric dynamics, understanding critical issues of climate change, improving the estimation of air pollutant dispersion, and creating better initial conditions (ICs) for numerical weather predictions (NWPs; WMO, 2017; Baker et al., 2014; Pu et al., 2017; Zhang and Pu, 2010; Pu et al. 2010; Rennie et al., 2021b). Large areas of the tropical atmosphere lack measurement of wind profiles, which suggests the potential for significantly improving forecasting skills for various tropical convective systems with additional wind observations (Baker et al., 2014). To provide high spatial



and temporal near-vertical wind profiles globally, the Aeolus satellite, the first spaceborne wind lidar, was launched by the European Space Agency (ESA) on 22 August 2018 (Reitebuch et al., 2020; ESA, 2022).

After successfully launching into a sun-synchronous orbit at about 320 km altitude with a weekly cycle, the Aeolus satellite now passes the equator at 18:00 (6:00) local time (LT) during ascending (descending) orbits (Andersson et al., 2008; Krisch
et al., 2022; ESA, 2022). The azimuth angle of the Aeolus satellite closely approaches $270°$ $(90°)$ for ascending (descending) orbits near the equator, and the viewing angle toward the atmosphere is $35°$ off-nadir. The horizontal line-of-sight (HLOS) wind component, derived from the measured wind along the laser beam line-of-sight (LOS), is approximately east-west oriented for most of the orbits (Krisch et al., 2022).

The Atmospheric Laser Doppler INstrument (ALADIN), on board the Aeolus mission, measures Doppler-shifted backscattered light scattering by atmospheric molecules and particulates with two separate interferometers: the Fizeau (Mie channel) for large particles, cloud droplets, ice crystals, dust, and aerosols, and the Fabry-Perot (Rayleigh channel) for moving molecules, including oxygen and nitrogen (Andersson et al., 2008; Reitebuch et al., 2009; Ingmann and Straume, 2016). Based on the signal-to-noise ratio, four types of HLOS winds are available, including Mie-clear, Mie-cloudy,
Rayleigh-clear, and Rayleigh-cloudy (Jos de Kloe et al., 2022). Rayleigh-clear winds perform better than Rayleigh-cloudy winds due to little or no contamination from Mie scattering. Mie-cloudy winds are better than Mie-clear winds because measurements from the Mie channel require strong backscattering from aerosols, water droplets, or ice crystals (Rennie et al., 2021a). The horizontal resolution of the Mie channel is typically about 10-15 km along the ground track and about 90 km for the Rayleigh channel. Vertically, the sizes of 24 range bins vary from 250 m to 2 km (Krisch et al., 2022).


Aeolus HLOS winds have been validated with different reference observations over many regions since launching, such as ground-based radar measurements (Zuo et al., 2022), ground-based coherent Doppler wind lidars (Wu et al., 2022), airborne Doppler wind lidars (Witschas et al., 2020; Witschas et al., 2022), in situ Atmospheric Motion Vectors (AMVs; Rani et al., 2022; Lukens et al., 2022), NWP equivalents (Rani et al., 2022; Martin et al., 2021), and radiosonde observations (Martin et
al., 2021; Baars et al., 2020). Validation shows that the quality of Mie-cloudy winds is better than that of Rayleigh-clear winds (Zuo et al., 2022; Rani et al., 2022). Furthermore, Aeolus HLOS winds can capture atmospheric dynamic structures well, such as the Low-Level Jet (LLJ), Tropical Easterly Jet (TEJ; Rani et al., 2022), atmospheric gravity waves (GWs; Banyard et al., 2021). As the Aeolus products are continuously calibrated and validated, the product processor is updated and the performance of the Aeolus Level-2B (L2B) wind product improves (Wu et al., 2022). Thus, the current Aeolus products
are suitable for data assimilation (DA) in the global forecast system (GFS; Pourret et al., 2022; Guo et al., 2021).

Assimilation of Aeolus HLOS winds has already been shown to improve analyses and forecasts in many global NWP models, including the Météo-France global NWP model ARPEGE (Pourret et al., 2022), the Canadian Global Deterministic



Prediction System (GDPS; Laroche and St-James, 2022), and ECMWF's Integrated Forecasting System (IFS; Rennie et al.,
2021a). Several other studies have assessed the impacts of assimilating Aeolus HLOS winds on tropical cyclones (TCs) by
Observing System Experiments (OSEs). Rani et al. (2022) conducted OSEs to examine the impacts of assimilation of Aeolus
HLOS winds on simulations of the location, intensity, and vertical extent of North Indian Ocean (NIO) cyclones, and the
improvements due to Aeolus winds were marginal. Marinescu et al. (2022) carried out OSEs with the operational Hurricane
Weather and Research Forecasting (HWRF) model to assess the impacts of assimilating Aeolus HLOS winds on forecasting
TCs, and their results suggest that the most significant potential impacts from assimilation of Aeolus observations are likely
to occur in the upper troposphere and lower stratosphere and within about 500 km of the TC center. The OSEs performed
with the NOAA Finite-Volume Cubed-Sphere Global Forecast System (FV3GFS) suggested that assimilation of Aeolus
HLOS winds can reduce track forecast error in the Eastern Pacific basin and Atlantic basin (Garrett et al., 2022). However,
the potential impacts of assimilating Aeolus near-real-time HLOS winds on simulations of TCs and tropical convective
systems have not been investigated. From August to September 2021, the NASA Convective Processes Experiment -
Aerosols & Winds (CPEX-AW) field campaign, in collaboration with ESA, conducted post-launch calibration and validation
activities for the Aeolus satellite near St. Croix over the Atlantic Ocean (AO), marking notable TCs and other tropical
convective systems observed by the Aeolus satellite. In this study, we use Hurricane Ida (2021), a major hurricane, and a
mesoscale convective system (MCS) embedded in an African easterly wave (AEW) during NASA's CPEX-AW campaign to
assess the impacts of assimilating Aeolus HLOS winds.

The paper is organized as follows. Section 2 introduces the mesoscale community Weather Research and Forecasting (WRF)
model, the NCEP Gridpoint Statistical Interpolation (GSI)-based three-dimensional ensemble-variational (3DEnVAR)
hybrid DA system, the Integrated Multi-satellitE Retrievals for GPM (IMERG) precipitation dataset, and the statistical
evaluation metrics. Section 3 discusses the impacts of assimilating Aeolus HLOS winds on forecasts of Hurricane Ida
(2021). Section 4 diagnoses the influence of assimilation of Aeolus HLOS winds on the analysis of atmospheric conditions
associated with Hurricane Ida (2021). Section 5 evaluates the influence of assimilation of Aeolus HLOS winds on an MCS
embedded in an AEW. Finally, Sect. 6 summarizes the results and assesses the need for future work.




## 2 WRF Model, DA system, IMERG, and statistical evaluation metrics

### 2.1 WRF Model

An advanced research version of WRF model (WRF-ARW) V4.4 (Skamarock et al., 2019) is applied in this study, with 61 terrain-following levels and the model top set at 10 hPa, as indicated by Table 1. Two domains are employed in all experiments. Their sizes differ in the experiments of Hurricane Ida (2021) and the MCS within an AEW, as illustrated in Fig. 1. The horizontal resolutions of the parent and inner domains are 12 and 4 km, and the time steps of the parent and inner domains are 30 and 10 s, respectively. The boundary conditions of the parent domain are obtained from the $0.25° × 0.25°$

horizontal resolution NCEP GFS final analyses (FNL). Details of the parameterization scheme options are listed in Table 1.

**Table 1: Configuration of the WRF-GSI cycling DA system**

| | | |
|---|---|---|
| Model | Resolution | 12 km (domain 1, d01, or parent domain), and 4 km (domain 2, d02, or inner domain) |
| | | 61 vertical levels with model top at 10 hPa |
| | Time steps | 30 s (d01) and 10 s (d02) |
| | Physics | Longwave radiation: RRTM (Mlawer et al., 1997) |
| | | Shortwave radiation: Dudhia (Dudhia, 1989) |
| | | Microphysics: WSM6 (Hong and Lim, 2006) |
| | | Cumulus: Kain-Fritsch (Activated only in the parent domain; Kain, 2004) |
| | | PBL: YSU (Hong et al., 2006) |
| | | Surface layer: Monin-Obukhov Similarity (Jiménez et al., 2012) |
| | | Land surface: Unified Noah LSM (Tewari et al., 2004) |
| | Boundary condition | Spin up and cycling DA period: NCEP GFS FNL |
| | | Forecast period: NCEP GFS Forecasts |
| Analysis | DA system | NCEP GSI-based 3DEnVAR hybrid DA system V3.7 |
| | State vector | $u$ (Zonal wind), $v$ (Meridional wind), $tv$ (Virtual temperature), $q$ (Specific humidity), $prse$ (Pressure), $ps$ (Surface pressure), $sst$ (Sea surface temperature) |
| | Control vector | $sf$ (Stream function; 0.60), $vp_{ub}$ (Unbalanced velocity potential; 0.60), $ps_{ub}$ (Unbalanced surface pressure; 0.75), $t_{ub}$ (Unbalanced temperature; 0.75), $rh$ (Normalized RH; 0.75), $sst$ (Sea surface temperature; 1.00) The numbers indicate normalized scale factors for their background error variances. |
| | Assimilation window | $\pm 3$ h |
| | Background error covariance | Static part (0.2), and flow-dependent part (0.8, estimated from 80-member 6-h ensemble forecasts from the NCEP GSI 4DEnVAR system) |
| | Ensemble localization scale | Horizontal: 110 km |
| | | Vertical: 3 grid units |

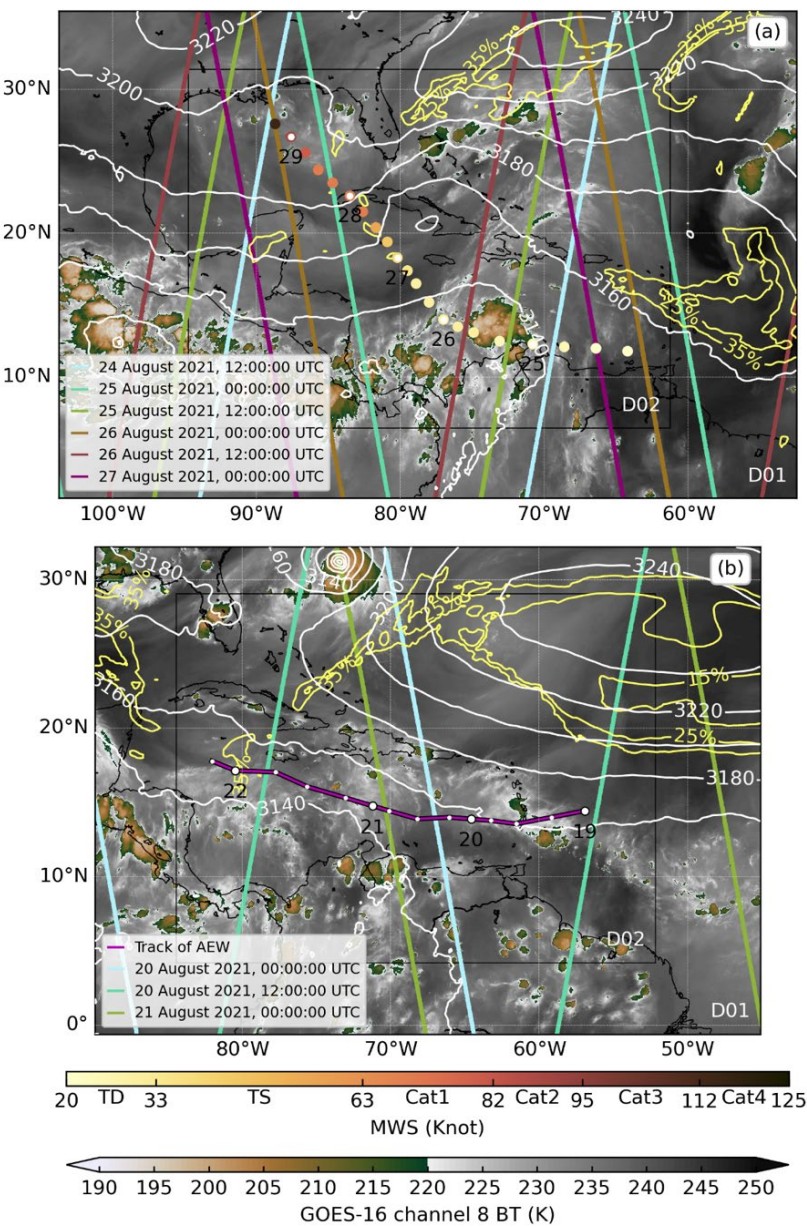

**Figure 1: Two nested domains (d01: 12 km, and d02: 4 km) in experiments 2418_L2B (a) and 1918_L2B (b). The Aeolus measurement swaths (colored line) are from 12 UTC 24 Aug to 00 UTC 27 Aug 2021 (a) and from 00 UTC 20 Aug to 00 UTC 21 Aug 2021 (b). The track and maximum surface wind speeds (MWSs, colored dot) of Hurricane Ida (2021) in (a) are adopted from the NHC best-track data, and the days are illustrated below the open markers indicating 00 UTC. Two classifications (TD: tropical depression; TS: tropical storm) and different categories (Cat1: Category 1; Cat2: Category 2; Cat3: Category 3; Cat4: Category 4) based on the Saffir-Simpson Hurricane wind scale are given below the color bar of MWS. The AEW in (b) is tracked manually using the GFS-analyzed 700-hPa relative vorticity maxima (purple line with black edges). The white dots with the numbers below along the AEW track indicate 00 UTC of a day. The 700-hPa geopotential height (white line) and 850-hPa relative humidity (RH; yellow line), obtained from GFS analysis, are overlaid atop GOES-16 channel 8 BTs (shaded) at 12 UTC 25 Aug (a), and 00 UTC 21 Aug 2021 (b). Only RH less than 35% is shown in (a) and (b).**





**2.2 NCEP GSI-based 3DEnVAR hybrid DA system**

The DA system used in this study is the NCEP GSI-based 3DEnVAR hybrid DA system V3.7, the details of which are listed in Table 1. The cost function of the GSI system has two terms: the background and observational error terms.

$$J(\boldsymbol{x}, \boldsymbol{\beta}) = \frac{1}{2}(\boldsymbol{x_b} - \boldsymbol{x})^T(\alpha_1 \mathbf{B_1} + \alpha_2 \mathbf{B_2})^{-1}(\boldsymbol{x_b} - \boldsymbol{x}) + \frac{1}{2}[\boldsymbol{y_0} - H(\boldsymbol{x})]^T \mathbf{R}^{-1}[\boldsymbol{y_0} - H(\boldsymbol{x})]$$

(1)

In the background error term (the first term in Eq. (1)), $\boldsymbol{x}$ is the analysis, and $\boldsymbol{x_b}$ is the first guess, namely a 6-h WRF model simulation. The hybrid background error covariance matrix consists of a prescribed static part $\mathbf{B_1}$, and a flow-dependent part $\mathbf{B_2}$. The weighting factors of these two portions are 0.2 ($\alpha_1$) and 0.8 ($\alpha_2$), respectively. The flow-dependent part is estimated by 80-member 6-h ensemble forecasts from the NCEP GSI 4DEnVAR system. The homogeneous isotropic horizontal ensemble localization scale is 110 km, and the vertical localization scale is 3 grid units (see Table 1). In the observational

error term (the second term in Eq. (1)), $\boldsymbol{y_0}$ is the observation, H is the forward model, and $\mathbf{R}$ is the observation error covariance. Thus, the innovation Observation-Minus-Background (OmB) is defined as $\boldsymbol{y_0} - H(\boldsymbol{x})$.

The observations $\boldsymbol{y_0}$ in this study are the Aeolus L2B baseline 12 near-real-time HLOS winds (ESA, 2021). The quality control (QC) steps used in this study are the following:

1.  Mie-clear and Rayleigh-cloudy products are not used in this study due to their poor quality (Rennie et al., 2021a; Pourret et al., 2022; Laroche and St-James, 2022).

2.  Aeolus HLOS winds marked as invalid retrievals by the L2B processor are eliminated (Rennie et al., 2021a).

3.  The GSI system rejects Mie-cloudy and Rayleigh-clear HLOS winds when their L2B estimated instrumental errors (before scaling) are over 12 and 10 m s$^{-1}$, respectively (Rennie et al., 2021a).

4.  Aeolus HLOS winds are discarded when their absolute value is larger than 1000 m s$^{-1}$.

5.  A gross check is adopted to remove observations with normalized OmBs (OmB normalized by $\sigma$) larger than $4\sigma$, while $\sigma$ is the standard deviation of the observation errors (similar to QC Step 6 in Pourret et al. (2022)).

The forward model H($\boldsymbol{x}$) of the Mie-cloudy and Rayleigh-clear HLOS winds (Pourret et al., 2022; Laroche and St-James,

2022; Rennie et al., 2021a) is defined as:

$$H(\boldsymbol{x}) = -\boldsymbol{u}\sin\theta - \boldsymbol{v}\cos\theta$$

(2)

The forward model H($\boldsymbol{x}$) interpolates the WRF model winds (zonal wind component $\boldsymbol{u}$ and meridional wind component $\boldsymbol{v}$) according to the geolocation of the Aeolus observations and projects the interpolated model winds to the HLOS winds with



the horizontal azimuth angle $\theta$. The vertical velocity is assumed to be zero in the forward model because it seems to be a second-order problem even for tropical convective systems (Rennie and Isaksen, 2020).

The observation error covariance matrix $\mathbf{R}$ can be divided into $\mathbf{R} = \Sigma^{\frac{1}{2}}\mathbf{C}\Sigma^{\frac{1}{2}}$. $\mathbf{C}$ is the correlation matrix. Since we assume that the observations are uncorrelated in this study, $\mathbf{C}$ is an identity matrix. $\Sigma^{\frac{1}{2}}$ is a diagonal matrix of the standard deviation of the

observation error $\sigma$. The observation error consists of the instrument and representation error, while the representation error includes the observation-operator error and the error due to unresolved scales and processes (Janjić et al., 2018). Since the representation error is unknown and the primary goal of this study is to investigate the impacts of assimilation of Aeolus HLOS winds on Hurricane Ida (2021) and an MCS embedded in an AEW, we simply use the L2B dynamic estimated instrumental errors to estimate the standard deviation of the observation error $\sigma$. Figure 2 shows scatter plots of the valid

Mie-cloudy (a) and Rayleigh-clear (b) HLOS winds against their estimated instrumental errors from 25 August 2021, 00:00:00 UTC to 26 August 2021, 18:00:00 UTC (cycling DA period of experiment 2418_L2B). The Mie-cloudy HLOS winds are clustered primarily between -20 and 20 m s$^{-1}$, and their estimated instrumental errors lie mostly between 1 and 4 m s$^{-1}$, so the standard deviation of the observation errors for the Mie-cloudy winds is simply assigned to 2.5 m s$^{-1}$ in this study, as indicated by the black line in Fig. 2 (a). The Rayleigh-clear HLOS winds are mainly between -30 and 30 m s$^{-1}$,

and their estimated instrumental errors vary primarily from 3 to 6 m s$^{-1}$, so the standard deviation of the observation errors for the Rayleigh-clear winds is simply set at 4.5 m s$^{-1}$. The Rayleigh-clear winds have a higher standard deviation of observation errors because the quality of the Mie-cloudy winds is better than that of the Rayleigh-clear winds (Zuo et al., 2022; Rani et al., 2022).

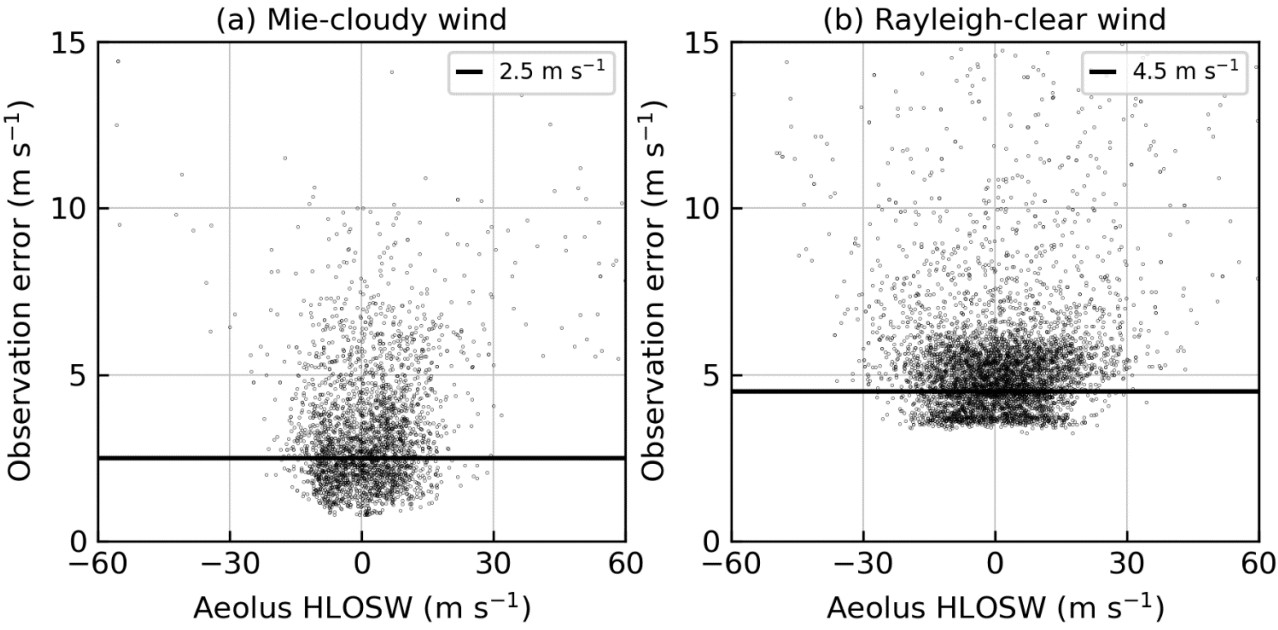


**Figure 2: Scatter plots (black circle) of valid Aeolus HLOS winds against their estimated instrumental errors from 00 UTC 25 Aug to 18 UTC 26 Aug 2021 (cycling DA period of experiment 2418_L2B) for Mie-cloudy winds (a) and Rayleigh-clear winds (b). The black lines in (a) and (b) indicate the estimated observation errors for the greatest population of valid Aeolus HLOS winds, which are 2.5 and 4.5 m s$^{-1}$, respectively. The results of experiments 2406_L2B, 2412_L2B, and 2500_L2B are similar to those of experiment 2418_L2B.**

## 2.3 IMERG Data

The precipitation dataset used for verifying rainfall forecasts in this study is IMERG Version 6B-Final. IMERG precipitation estimates combine various satellite passive microwave sensors related to precipitation, including the GPM constellation and microwave precipitation-calibrated geo-IR fields. IMERG precipitation estimates are half-hourly and on a $0.1° \times 0.1°$ grid over the globe. This dataset is available from June 2000 to the present (Huffman et al., 2019).

## 2.4 Statistical evaluation metrics

### 2.4.1 RMSE

Root-mean-square error (RMSE) can measure errors of the track, MWS, and minimum sea-level pressure (MSLP) between the forecasts and NHC best-track data. RMSE is defined as follows:

$$RMSE = \sqrt{\left(x - x_{ref}\right)^2}$$

(3)

where $x$ is the location, MWS, or MSLP forecast of a TC, while $x_{ref}$ is adopted from the NHC best-track data.

### 2.4.2 ETS

An Equitable Threat Score (ETS) is a corrected ratio of the number of correctly predicted events to the total number of predicted or observed events. The definition of ETS is:

$$ETS = \frac{N_H - ref}{N_H + N_{FA} + N_M - ref}$$

(4)

where $ref$ is the chance forecast:

$$ref = \frac{(N_H + N_{FA})(N_H + N_M)}{N_H + N_{FA} + N_H + N_M}$$

(5)

Other variables are computed by a contingency table (Table 2), which defines Hit ($N_H$), False Alarm ($N_{FA}$), Miss ($N_M$), and Correct Negative ($N_{CN}$).






**Table 2: Contingency table**

|  |  | Observation | |
|---|---|---|---|
|  |  | Yes | No |
| Forecast | Yes | Hit ($N_H$) | False Alarm ($N_{FA}$) |
|  | No | Miss ($N_M$) | Correct Negative ($N_{CN}$) |

## 3 Impacts of Aeolus data on numerical simulations of Hurricane Ida (2021)

### 3.1 Case description and experiment design

Ida (2021) originated from an AEW on 14 August 2021. On 24 August 2021, the AEW moved into the Caribbean Sea, reached the area near Aruba, Bonaire, and Curaçao, and interacted with MCSs along the northern coast of South America.

On 25 August 2021, 12:00:00 UTC, as shown by Fig. 1 (a), the convection, indicated by the GOES-R channel 8 brightness temperatures (BTs), was concentrated in the eastern area of a broad low-pressure system south of the subtropical ridge over the southwestern Caribbean Sea, which is shown by 700-hPa geopotential height and 850-hPa RH. Steered by the subtropical ridge, the convective disturbance began to turn north to northwestward on 26 August 2021, as demonstrated by the track of Ida (2021) in Fig. 1 (a). A TD formed on 26 August 2021, 12:00:00 UTC, and then intensified into a TS 6 hours after

cyclogenesis. Early on 27 August 2021, the first rapid intensification (RI) period occurred, and Ida (2021) strengthened into a hurricane on 27 August 2021, 18:00:00 UTC. After moving across the Isle of Youth, Ida (2021) made landfall in Cuba around 28 August 2021, 00:00:00 UTC. After passing Cuba, Ida (2021) experienced the second RI period from 28 August 2021, 12:00:00 UTC to 29 August 2021, 12:00:00 UTC (Beven II et al., 2022).

As shown by Table 3, 8 experiments, including experiments 2406, 2406_L2B, 2412, 2412_L2B, 2418, 2418_L2B, 2500, and 2500_L2B, are performed to investigate the impacts of assimilating Aeolus Mie-cloudy and Rayleigh-clear HLOS winds on the analysis and subsequent forecasts of Hurricane Ida (2021). Experiment 2406 is initialized by the NCEP GFS FNL on 24 August 2021, 06:00:00 UTC. After a 6-h spin-up, a cycling DA period from 24 August 2021, 12:00:00 UTC to 26 August 2021, 06:00:00 UTC (8 DA cycles in total) is performed in experiment 2406. Only NCEP ADP global upper air and surface

weather observations are assimilated. The subsequent 48-hour forecasts are initialized from the analysis of DA cycles 5, 6, 7, and 8. Experiment 2406_L2B is similar to experiment 2406, but it also assimilates Aeolus HLOS winds during cycling DA. The forecasts are not initialized from the analysis of DA cycles 1, 2, 3, and 4 in this study because the impacts of assimilating Aeolus HLOS winds are tiny if the number of DA cycles is not large enough. Since Ida (2021) was just a broad low-pressure system interacting with MCSs along the northern coast of South America on 24 August 2021, the forecasts of

Ida (2021) are sensitive to its ICs. Consequently, except for experiments 2406 and 2406_L2B, we carry out three more sets of experiments with different initial times: 2412 and 2412_L2B (initialized on 24 August 2021, 12:00:00 UTC), 2418 and



2418_L2B (initialized on 24 August 2021, 18:00:00 UTC), and 2500 and 2500_L2B (initialized on 25 August 2021, 00:00:00 UTC). The cycling DA periods of all experiments are before the first RI of Ida (2021), enabling us to investigate the impacts of assimilating Aeolus HLOS winds on the intensity forecasts of the first and second rapid RI periods of Ida (2021).

**Table 3: List of experiment configurations**

| Experiment | Case | Initial Time | Cycling DA Period | DA Observation | Forecast |
|---|---|---|---|---|---|
| 2406 | Ida (2021) | 24 August 2021, 06:00:00 UTC | From 24 August 2021, 12:00:00 UTC to 26 August 2021, 06:00:00 UTC (8 cycles) | NCEP conventional observations | 48-h forecasts initialized from DA Cycle 5, 6, 7, and 8 |
| 2406_L2B | Ida (2021) | 24 August 2021, 06:00:00 UTC | From 24 August 2021, 12:00:00 UTC to 26 August 2021, 06:00:00 UTC (8 cycles) | NCEP conventional observations and Aeolus L2B HLOS winds | 48-h forecasts initialized from DA Cycle 5, 6, 7, and 8 |
| 2412 | Ida (2021) | 24 August 2021, 12:00:00 UTC | From 24 August 2021, 18:00:00 UTC to 26 August 2021, 12:00:00 UTC (8 cycles) | NCEP conventional observations | 48-h forecasts initialized from DA Cycle 5, 6, 7, and 8 |
| 2412_L2B | Ida (2021) | 24 August 2021, 12:00:00 UTC | From 24 August 2021, 18:00:00 UTC to 26 August 2021, 12:00:00 UTC (8 cycles) | NCEP conventional observations and Aeolus L2B HLOS winds | 48-h forecasts initialized from DA Cycle 5, 6, 7, and 8 |
| 2418 | Ida (2021) | 24 August 2021, 18:00:00 UTC | From 25 August 2021, 00:00:00 UTC to 26 August 2021, 18:00:00 UTC (8 cycles) | NCEP conventional observations | 48-h forecasts initialized from DA Cycle 5, 6, 7, and 8 |
| 2418_L2B | Ida (2021) | 24 August 2021, 18:00:00 UTC | From 25 August 2021, 00:00:00 UTC to 26 August 2021, 18:00:00 UTC (8 cycles) | NCEP conventional observations and Aeolus L2B HLOS winds | 48-h forecasts initialized from DA Cycle 5, 6, 7, and 8 |
| 2500 | Ida (2021) | 25 August 2021, 00:00:00 UTC | From 25 August 2021, 06:00:00 UTC to 27 August 2021, 00:00:00 UTC (8 cycles) | NCEP conventional observations | 48-h forecasts initialized from DA Cycle 5, 6, 7, and 8 |
| 2500_L2B | Ida (2021) | 25 August 2021, 00:00:00 UTC | From 25 August 2021, 06:00:00 UTC to 27 August 2021, 00:00:00 UTC (8 cycles) | NCEP conventional observations and Aeolus L2B HLOS winds | 48-h forecasts initialized from DA Cycle 5, 6, 7, and 8 |
| 1918 | MCS | 19 August 2021, 18:00:00 UTC | From 20 August 2021, 00:00:00 UTC to 21 August 2021, 00:00:00 UTC (5 cycles) | NCEP conventional observations | 30-h forecasts initialized from DA Cycle 5 |
| 1918_L2B | MCS | 19 August 2021, 18:00:00 UTC | From 20 August 2021, 00:00:00 UTC to 21 August 2021, 00:00:00 UTC (5 cycles) | NCEP conventional observations and Aeolus L2B HLOS winds | 30-h forecasts initialized from DA Cycle 5 |

**3.2 Track forecasts**

Figure 3 (a) compares the 48-hour track forecasts between experiments 2406 (without assimilation of Aeolus HLOS winds) and 2406_L2B (with assimilation of Aeolus HLOS winds) during the last four cycles of the cycling DA period (from 25 August 2021, 12:00:00 UTC to 26 August 2021, 06:00:00 UTC). The 48-hour track forecasts of experiment 2406 initialized from the last four DA cycles show that all track forecasts, where Ida generally moves northwestward, are southwest of the NHC best track. As shown by Fig. 3 (b), (c), and (d), the 48-hour track forecasts of experiments 2412, 2418, and 2500 (as in experiment 2406) also have systematic biases toward the southwest of the NHC best track of Ida (2021). In addition, the track forecast errors are reduced in experiments without assimilation of Aeolus HLOS winds (see the numbers in Fig. 3 (e)-(h)) when more NCEP conventional data are assimilated, in addition to the forecasts initialized from DA cycles 5 and 6 of experiment 2406. Compared to experiments without assimilation of Aeolus HLOS winds, the 48-hour track forecasts become closer to the NHC best track of Ida (see Fig. 3 (a)-(d)), and the averaged track forecast errors are reduced consistently in the experiments with assimilation of Aeolus HLOS winds, as shown by Fig. 3 (e)-(h). The reductions of the averaged track forecast errors range from 10 to around 60 km after assimilation of Aeolus HLOS winds.







**Figure 3: Comparison of 48-h track forecasts of Hurricane Ida (2021) initialized from the analysis of DA cycles 5 (blue), 6 (green), 7 (brown), and 8 (pink) between experiments 2406 (dashed lines) and 2406_L2B (solid lines) (a), 2412 and 2412_L2B (b), 2418 and 2418_L2B (c), and 2500 and 2500_L2B (d). Tracks of Ida (2021) adopted from the NHC best-track data are shown by solid black lines in (a)-(d), and the days are illustrated above the open markers indicating 00 UTC. The forecasts of Ida (2021) are tracked by**
**the GFDL vortex tracker. Compared with the experiments without assimilation of Aeolus HLOS winds, the averaged RMSE reductions of the 48-h track forecasts initialized from the analysis of DA cycles 5 (blue), 6 (green), 7 (brown), and 8 (pink) for experiments 2406_L2B, 2412_L2B, 2418_L2B, and 2500_L2B are shown in (e)-(h), respectively. The numbers indicate the averaged RMSEs of track forecasts of the reference experiments. Positive values of RMSE reductions mean improvement, while negative values indicate degradation.**

**3.3 Intensity forecasts**

As introduced in Sect. 3a, Ida (2021) made cyclogenesis on 26 August 2021, 12:00:00 UTC and strengthened into a TS 6 hours later. Ida (2021) experienced its first RI period from 00:00:00 UTC to 18:00:00 UTC on 27 August 2021. After making landfall and passing Cuba, Ida (2021) experienced the second RI period from 28 August 2021, 12:00:00 UTC to 29 August 2021, 12:00:00 UTC. Figure 4 (a) compares the 48-hour MSLP forecasts initialized from the last four cycles (from

25 August 2021, 12:00:00 UTC to 26 August 2021, 06:00:00 UTC) between experiments 2406 (without assimilation of Aeolus HLOS winds) and 2406_L2B (with assimilation of Aeolus HLOS winds). The 48-hour MSLP forecasts of experiments 2406 and 2406_L2B capture the intensification processes well, although they cannot capture the platform between the two RI periods (from 27 August 2021, 18:00:00 UTC to 28 August 2021, 12:00:00 UTC) because their forecasts fail to predict the landfall of Ida (2021) in Cuba (see Fig. 3 (a)). Figure 4 (e) indicates a neutral impact of assimilation of

Aeolus HLOS winds on the MSLP forecasts for Ida (2021), compared to experiment 2406. Figure 5 (a) and (e) compare the 48-hour MWS forecasts initialized from the last four cycles between experiments 2406 and 2406_L2B, and the conclusions are consistent with the MSLP forecasts of experiments 2406 and 2406_L2B. Similarly, the 48-h forecasts of experiments 2412 and 2412_L2B also predict the MSLP (see Fig. 4 (b)) and the MWS (see Fig. 5 (b)) well, expect for the forecasts initialized from DA cycle 5, and the impacts of assimilating Aeolus HLOS winds are neutral as well (see Fig. 4 (f)).


**Figure 4: As in Fig. 3, but for MSLP forecasts.**



**Figure 5: As in Fig. 3, but for MWS forecasts.**



Nevertheless, Fig. 4 and Fig. 5 (g) and (h) indicate the positive impacts of assimilating Aeolus HLOS winds on the MSLP

and MWS forecasts, compared to experiments 2418 and 2500. The averaged improvements of the MSLP forecasts are up to

4 hPa, while the averaged improvements of the MWS forecasts reach 5 Knots, after assimilation of Aeolus HLOS winds. As

shown by Fig. 4 and Fig. 5 (c) and (d), the 48-h MSLP and MWS forecasts of experiments 2418 and 2500 underestimate the

intensity of Ida (2021) from the first to second RI period. The corresponding 48-h MSLP and MWS forecasts from

experiments 2418_L2B and 2500_L2B also underestimate the intensity of Ida, but they are stronger than those of the

reference experiments owing to deeper MSLPs and higher MWSs.

**3.4 Precipitation forecasts**

To evaluate the impacts of assimilating Aeolus HLOS winds on the rainfall structures of Ida (2021), we calculate ETS scores

using the forecasts of 6-h accumulated precipitation against the IMERG precipitation data within a $10° \times 10°$ box centered

on Hurricane Ida (2021). Figure 6 (a) exhibits the reductions of the 48-h averaged ETS scores of experiment 2406_L2B

(with assimilation of Aeolus HLOS winds) for different precipitation thresholds when compared to experiment 2406

(without assimilation of Aeolus HLOS winds). The improvements of experiment 2406_L2B are evident for 15 mm

precipitation thresholds, compared with experiment 2406. Positive impacts of assimilation of Aeolus HLOS winds on the 6-h

accumulated precipitation forecasts for 10 mm and 15 mm precipitation thresholds can also be found in experiments

2412_L2B, 2418_L2B, and 2500_L2B when compared to their corresponding reference experiments. These results imply

that assimilating Aeolus HLOS winds can improve simulations of the rainfall structure of strong convection for Ida (2021).





**Figure 6: Reductions of the 48-h averaged ETSs of forecasts initialized from the analysis DA cycles 5, 6, 7, and 8 for different precipitation thresholds: 10 mm (green) and 15 mm (pink) in experiments 2406_L2B (a), 2412_L2B (b), 2418_L2B (c), and 2500_L2B (d), compared with the corresponding experiments without assimilation of Aeolus HLOS winds. The ETS scores are calculated using the forecasts of 6-h accumulated precipitation against the IMERG precipitation data within a $10° × 10°$ box centered on Hurricane Ida (2021). Positive values of ETS score reductions mean improvement, while negative values indicate degradation.**



## 4 Diagnosis of influence of Aeolus DA on analysis of Hurricane Ida (2021)

### 4.1 Analysis increments of Aeolus HLOS winds

As mentioned in Sect. 3c, assimilating Aeolus HLOS winds leads to better intensity forecasts in experiments 2418_L2B and 2500_L2B from the first to second RI periods of Ida (2021). Thus, it is necessary to understand how assimilation of Aeolus HLOS winds influences Ida's dynamic and thermodynamic structure and then improves the intensity forecasts in experiments 2418_L2B and 2500_L2B. Commonly, improvements in a hurricane's inner core structure lead to positive impacts on hurricane intensity forecasts. Therefore, we choose experiments that assimilate Aeolus measurement swaths close

to the center of Ida (2021) to examine their analysis increments over the hurricane's inner core region. As demonstrated by Fig. 1 (a), the Aeolus descending measurement swath on 25 August 2021, 12:00:00 UTC, and another on 26 August 2021, 12:00:00 UTC, are close to the center of Ida (2021) during the cycling DA period of experiments 2418_L2B and 2500_L2B and are suitable for investigating the analysis increments of Aeolus HLOS winds. Figure 7 (a) and (e) show vertical cross sections of the analysis increments of Mie-cloudy winds and Rayleigh-clear winds for the selected Aeolus measurement

swath on 25 August 2021, 12:00:00 UTC in experiment 2418_L2B. Since the HLOS winds of the Mie channel are derived by the Doppler-shifted backscattered light from the Fizeau interferometer, which detects aerosols and small hydrometeors, the analysis increments of Mie-cloudy winds are located primarily near the cloud top (between 10 and 16 km), as revealed by Fig. 7 (a). The analysis increments of Mie-cloudy winds are mostly positive near the center of Ida (2021). Figure 7 (b) shows that the Rayleigh-clear winds, whose range bin thickness is 750 m between 12 and 15 km, are measured from the

surface up to over 20 km in the tropical region ($0 - 30°$ N). In comparison, the maximum measurement height in the extratropical region ($30 - 60°$ N) is approximately 17.5 km, and the vertical resolution between 5 and 10 km (500 m) is higher due to the detection of the jet stream (ESA, 2020). However, ALADIN is totally attenuated by optically thick clouds or aerosols, so there are no Rayleigh-clear winds under the cloud top near the center of Ida (2021). The resolution of the Mie-cloudy winds is much finer than that of the Rayleigh-clear winds, but the Rayleigh-clear winds have more extensive

coverage. The analysis increments of Rayleigh-clear winds are roughly consistent with those of Mie-cloudy winds where they overlap. Figure 7 (b)-(d) and (f)-(h) are similar to Fig. 7 (a) and (e), respectively, but for different experiments or times. Note that the analysis increments of Mie-cloudy winds are mainly negative near the center of Ida (2021) on 25 August 2021, 12:00:00 UTC in experiment 2500_L2B.



**Figure 7: Vertical cross sections of analysis increments of Mie-cloudy winds for the Aeolus descending measurement swath close to the center of Ida (2021) in experiments 2418_L2B (a) and 2500_L2B (b) on 25 August 2021, 12:00:00 UTC. The triangles indicate the locations of Ida (2021) in the simulations on 25 August 2021, 12:00:00 UTC. (c)-(d) As in (a)-(b) but for 12 UTC on 26 August 2021. (e)-(h) As in (a)-(d), but for Rayleigh-clear winds.**


## 4.2 Vertical cross-section of analysis increments of zonal winds and RH

The Aeolus HLOS winds are perpendicular to the Aeolus orbit, about 10 degrees off the zonal direction (Krisch et al., 2022). Thus, assimilating Aeolus HLOS winds generally impact the zonal wind component more than the meridional wind component. Figure 8 (c) compares vertical cross sections of zonal wind increments along the selected Aeolus descending measurement swath on 25 August 2021, 12:00:00 UTC, between experiments 2418 and 2418_L2B. The significant impacts of assimilating Aeolus HLOS winds on the analysis increments of zonal winds are located at the upper troposphere (near the

cloud top), especially near the center of Ida (2021), which is consistent with Marinescu et al. (2022), and Garrett et al. (2022). As shown by Fig. 8 (c), the difference in the analysis increments of the zonal winds is negative at the cloud tops near the center of Ida (2021) due to the positive analysis increments of Mie-cloudy and Rayleigh-clear winds (descending orbit). Figure 8 (f) is similar to Fig. 8 (c), but for experiments initialized on 25 August 2021, 00:00:00 UTC. Figure 8 (f) demonstrates that the difference in the zonal wind increments is primarily positive at the cloud tops near the center of Ida

(2021), owing to the negative analysis increments of Mie-cloudy and Rayleigh-clear winds. Figure 9 examines vertical cross sections of analysis increments of RH along the selected Aeolus descending measurement swath on 25 August 2021, 12:00:00 UTC, for experiments initialized on 24 August 2021, 18:00:00 UTC, and 25 August 2021, 00:00:00 UTC. As with the zonal wind increments, the impacts of assimilating Aeolus HLOS winds on RH increments are also located primarily in the upper troposphere (between 10 and 15 km).

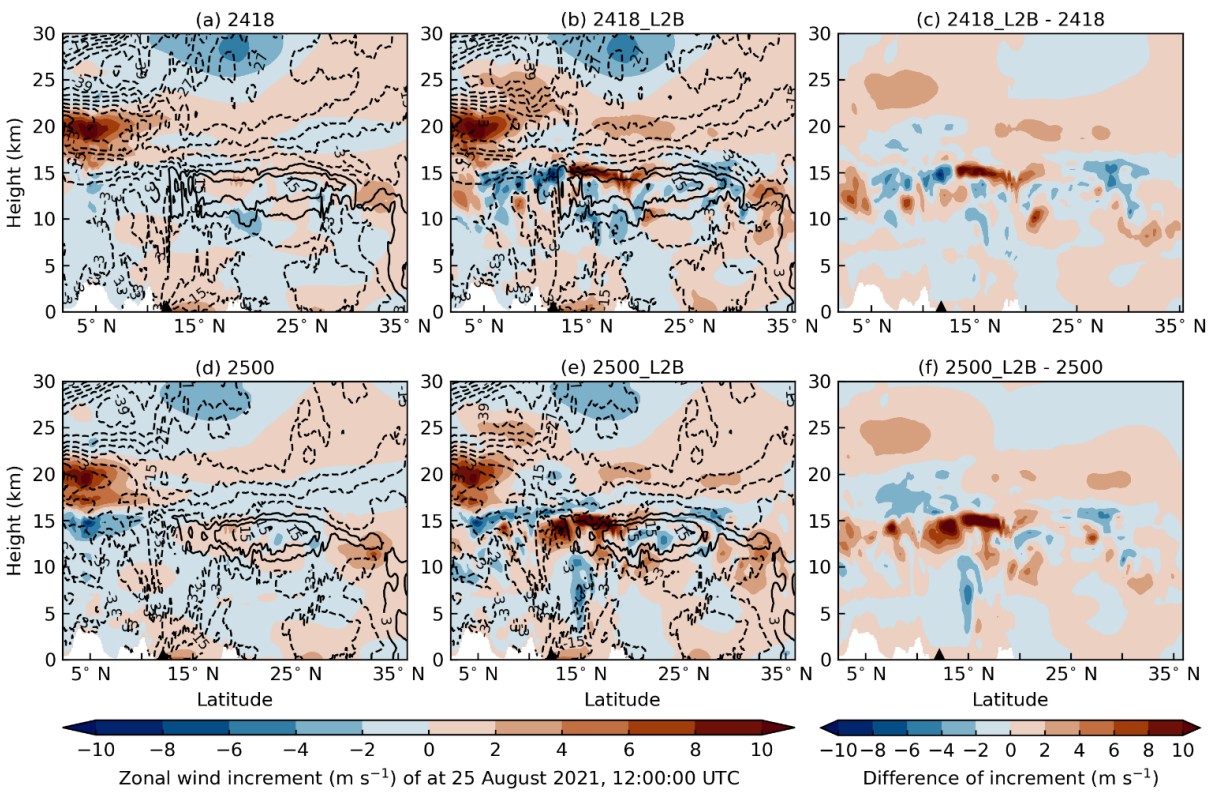




**Figure 8: Vertical cross section of zonal wind increments along the Aeolus descending measurement swath close to the center of Ida (2021) on 25 August 2021, 12:00:00 UTC in experiments 2418 (a) and 2418_L2B (b). (c) shows the differences in the zonal wind increments between experiments 2418_L2B and 2418. The triangles indicate the locations of Ida (2021) in the simulations. (d)-(f) As in (a)-(c), but for experiments initialized on 25 August 2021, 00:00:00 UTC.**

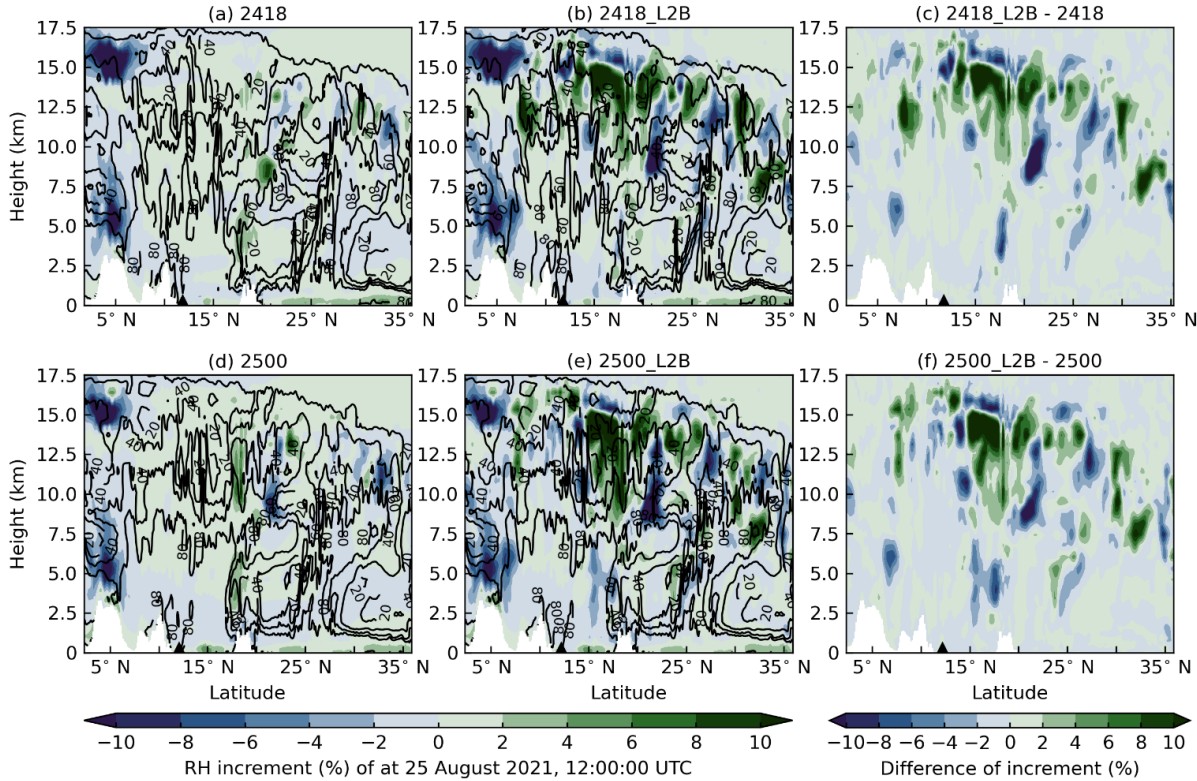

**Figure 9: As in Fig. 8, but for RH.**

## 4.3 Vertical profiles of averaged analysis increments of divergence and RH

To further investigate the impacts of assimilating Aeolus HLOS winds on the dynamic and thermodynamic structure of Hurricane Ida (2021), we calculate the averaged analysis increments of divergence and RH within a 300 km radius of the center of Ida (2021). Figure 10 (a) shows that after assimilation of Aeolus HLOS winds on 25 August 2021, 12:00:00 UTC, the analysis increments of divergence slightly increase in the upper level of the troposphere (near 200 hPa), which may help the intensification processes of Ida (2021). The differences in the analysis increments of divergence are tiny in the middle and lower troposphere between experiments 2418 and 2418_L2B because the major impacts of assimilation of Aeolus HLOS winds are located at the cloud top near the center of Ida, as mentioned in Sect. 3f. On 26 August 2021, 12:00:00 UTC, the analysis increments of divergence decrease in the lower and middle troposphere (below 400 hPa), which may be caused by assimilation of NCEP conventional observations and different center locations of Ida (2021) in experiments 2418 and 2418_L2B. Figure 10 (b) demonstrates that the analysis increments of divergence near 200 hPa become stronger on 25





August 2021, 12:00:00 UTC, and the analysis increments of convergence in the upper troposphere become weaker on 26
August 2021, 12:00:00 UTC after assimilation of Aeolus HLOS winds. Figure 10 (c) and (d) show that assimilation of
Aeolus HLOS winds has only tiny impacts on the moisture structure of Ida (2021). In short, assimilation of Aeolus HLOS
winds leads to stronger divergence in the upper level of the troposphere, which could be one reason for the improved
intensity forecasts of experiments 2418_L2B and 2500_L2B.

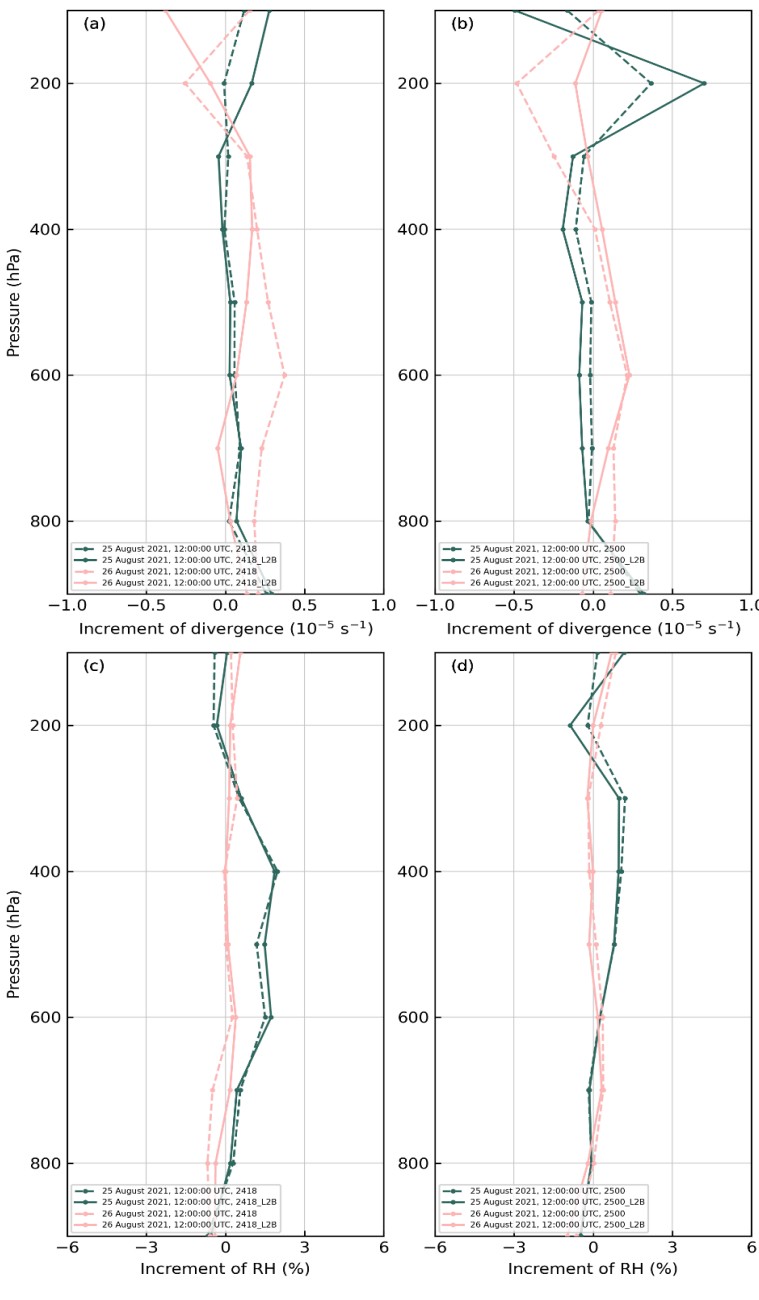





**Figure 10: Vertical profiles of averaged analysis increments of divergence (a) of experiments 2418 (dashed line) and 2418_L2B (solid line) on 25 August 2021, 12:00:00 UTC (green) and 26 August 2021, 12:00:00 UTC (pink). The analysis increments of divergence are averaged within a 300 km radius of the center of Ida (2021). (b) As in (a), but for experiments 2500 and 2500_L2B. (c)-(d) As in (a)-(b), but for averaged analysis increments of RH.**

## 5 Impacts of Aeolus data on numerical simulations of an MCS associated with an AEW

### 5.1 Case description and experiment design

As indicated by the GOES-R channel 8 (see Fig. 1 (b)), there was a strong subtropical high over the AO on 21 August 2021, 00:00:00 UTC. TS Henri, moving northeastward, was west of the subtropical high, and MCSs could be found on the north coast of South America. As shown by the track of GFS-analyzed 700-hPa relative vorticity maxima, an AEW, propagating westward on the south edge of the subtropical high, moved into the Caribbean Sea around 19 August 2021, 12:00:00 UTC. Steered by the subtropical ridge, the AEW, with scattered convection embedded inside, continued to move westward to

northwestward and reached the western Caribbean Sea around 22 August 2021, 00:00:00 UTC. Experiments 1918 and 1918_L2B are carried out to assess the impacts of the assimilation of Aeolus HLOS winds on the analysis and forecasts of the MCS associated with the AEW. Experiment 1918 is initialized by the NCEP GFS FNL on 19 August 2021, 18:00:00 UTC, and assimilates only the NCEP conventional observations with 3DVAR during the cycling DA period from 20 August 2021, 00:00:00 UTC to 21 August 2021, 00:00:00 UTC (5 DA cycles in total). The subsequent forecast is initialized from

the analysis of DA cycle 5. Experiment 1918_L2B is similar to experiment 1918 but also assimilates Aeolus HLOS winds.

### 5.2 Results

Figure 11 shows the 6-h accumulated precipitation from 21 August 2021, 18:00:00 UTC to 22 August 2021, 00:00:00 UTC, and the 850-hPa divergence over $5 \times 10^{-5} \ s^{-1}$ on 22 August 2021, 00:00:00 UTC. As shown by Fig. 11 (a), a large area of 6-h accumulated rainfall over 10 mm from the IMERG precipitation dataset is located within a 150 km radius and west of

the center of the AEW. Due to the downward airflow created by the heavy rainfall, divergence over $5 \times 10^{-5} \ s^{-1}$ from the GFS analysis can also be found west of the center of the AEW on 22 August 2021, 00:00:00 UTC. Figure 11 (b) shows only scattered convection and rainfall-induced low-level divergence near the center of the AEW in experiment 1918. As for experiment 1918_L2B (see Fig. 11 (c)), a large area of 6-h accumulated rainfall over 10 mm and rainfall-induced low-level divergence are located at the center of the AEW within a 150 km radius, which is more consistent with the IMERG

precipitation and the low-level divergence from the GFS analysis, implying a positive impact on numerical simulations of the MCS by assimilating Aeolus HLOS winds.



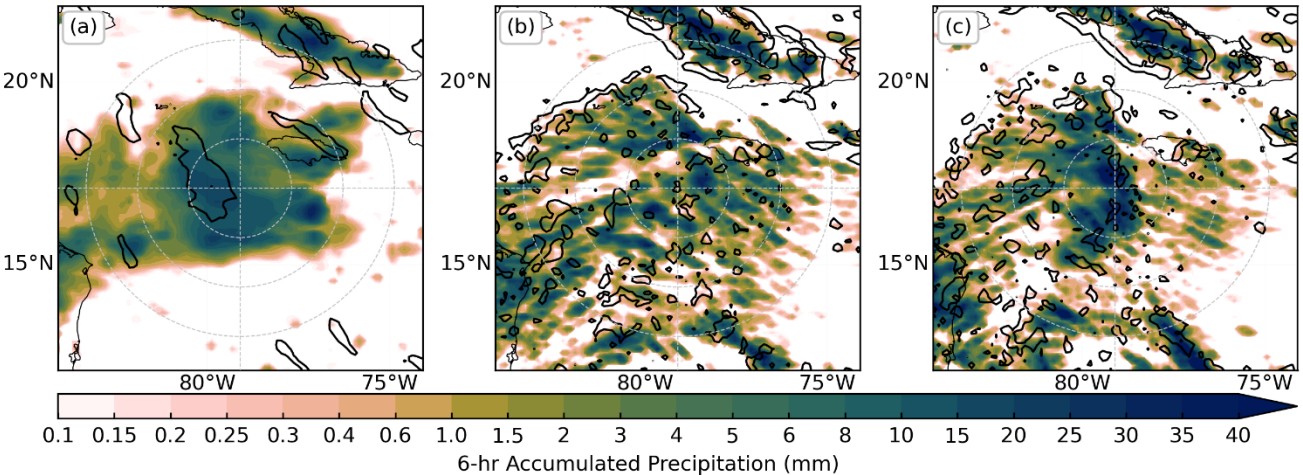

**Figure 11: (a) 6-h accumulated IMERG precipitation (color shading) from 21 August 2021, 18:00:00 UTC to 22 August 2021, 00:00:00 UTC and 850-hPa divergence over $5 \times 10^{-5}\ s^{-1}$ (black contour) from the GFS analysis on 22 August 2021, 00:00:00 UTC. (b)-(c) As in (a), but the precipitation and divergence are calculated by the WRF model simulations of experiment 1918 (b) and experiment 1918_L2B (c). The gray circles indicate radii of 150, 300, and 450 km, respectively.**

## 6 Conclusions

Measurement of three-dimensional wind profiles in the GOS is essential for improving the ICs of global NWPs, but it is insufficient over the oceans and remote land areas (WMO, 2017; Baker et al., 2014; Pu et al., 2017; Zhang and Pu, 2010; Pu et al. 2010; Rennie et al., 2021b). To fill the gap, the Aeolus satellite was launched by the ESA on 22 August 2018 and became the first spaceborne wind lidar (Reitebuch et al., 2020; ESA, 2022). Four types of Aeolus HLOS winds are available: Mie-clear, Mie-cloudy, Rayleigh-clear, and Rayleigh-cloudy (Jos de Kloe et al., 2022), but Mie-cloudy and Rayleigh-clear winds have better quality than the others (Zuo et al., 2022; Rani et al., 2022). The influences of assimilating Aeolus Mie-cloudy and Rayleigh-clear near-real-time HLOS winds on the forecasts of TCs and tropical convective systems have yet to be investigated. Thus, this study assesses the impacts of assimilating Aeolus Mie-cloudy and Rayleigh-clear HLOS winds on the analysis and forecasts of Hurricane Ida (2021) and an MCS embedded in an AEW during the CPEX-AW field campaign (2021). The WRF model and NCEP GSI-based 3DEnVAR hybrid DA system are applied in this study.

Ida (2021) originated from an AEW on 14 August 2021 and became a TD on 26 August 2021, 12:00:00 UTC. Six hours after cyclogenesis, it intensified into a TS. Ida (2021) experienced two RI periods: from 00 UTC to 18 UTC on 27 August 2021 (RI1); and from 28 August 2021, 12:00:00 UTC to 29 August 2021, 12:00:00 UTC (RI2). Between these two RI periods, Ida (2021) made landfall in Cuba around 28 August 2021, 00:00:00 UTC (Beven II et al., 2022). Eight experiments, including experiments 2406, 2406_L2B, 2412, 2412_L2B, 2418, 2418_L2B, 2500, and 2500_L2B, are carried out to investigate the impacts of assimilating Aeolus HLOS winds on the track and intensity forecasts of Ida (2021), especially during the two RI periods. Compared with the experiments without assimilation of Aeolus HLOS winds (experiments 2406, 2412, 2418, and





2500), the track forecasts are improved after assimilation of Aeolus HLOS winds (experiments 2406_L2B, 2412_L2B, 2418_L2B, 2500_L2B). The impacts on intensity forecasts are neutral in experiments 2406_L2B and 2412_L2B, while small but continuous improvements in intensity forecasts can be found in experiments 2418_L2B, and 2500_L2B. In addition, the ETS scores against the IMERG precipitation data near the center of Ida (2021) show that assimilating Aeolus HLOS winds

can improve the 6-h accumulated precipitation forecasts for strong convection (10 mm and 15 mm). One reason for the improved intensity and precipitation forecasts after assimilation of Aeolus HLOS winds is the stronger divergence in the upper level of the troposphere, as indicated by the averaged analysis increments of divergence within a 300 km radius of the center of Ida (2021) on 25 August 2021, 12:00:00 UTC and 26 August 2021, 12:00:00 UTC in experiments 2418_L2B and 2500_L2B. In addition to Hurricane Ida (2021), we also perform experiments 1918 and 1918_L2B to examine the impacts of

assimilating Aeolus HLOS winds on an MCS embedded in an AEW, which was steered by a subtropical high and propagated westward from 19 August 2021, 12:00:00 UTC to 20 August 2021, 00:00:00 UTC in the Caribbean Sea. The results of  the MCS demonstrate that assimilating Aeolus HLOS winds leads to better structure of the 6-h accumulated precipitation from 21 August 2021, 18:00:00 UTC to 22 August 2021, 00:00:00 UTC, and rainfall-induced low-level divergence on 22 August 2021, 00:00:00 UTC.


Although this study demonstrates positive impacts of assimilating Aeolus Mie-cloudy and Rayleigh-clear winds on the forecasts of Hurricane Ida (2021) and the MCS associated with an AEW, the assimilation techniques require further investigation and enhancement. More case studies are needed if the operational assimilation of Aeolus HLOS winds is required. Future studies will improve the technique of Aeolus HLOS wind assimilation with more cases of tropical

convective systems.

*Data availability.* The NCEP operational Global Forecast System analysis and forecasts at a 6-hourly interval are obtained from https://rda.ucar.edu/datasets/ds084.1 (last access: 31 August 2022, National Center for Atmospheric Research). The Aeolus L2B scientific wind products are accessed via https://aeolus-ds.eo.esa.int/oads/access/collection (last access: 05

September 2022, European Space Agency). The NHC best-track data can be accessed via https://www.nhc.noaa.gov/data (last access: 14 September 2022, National Hurricane Center and Central Pacific Hurricane Center). IMERG precipitation data are downloaded from https://disc.gsfc.nasa.gov/datasets/GPM_3IMERGHH_06/summary (last access: 12 April 2022, Goddard Earth Sciences Data and Information Services Center).

*Author contribution.* Chengfeng Feng collected the data, performed the experiments, analyzed the experiment results, and drafted the manuscript. Zhaoxia Pu evolved the overall research goals and aims, acquired the financial support for the project leading to this publication, and revised the manuscript critically.

*Competing interests.* The authors declare that they have no conflict of interest.




***Acknowledgements.*** We appreciate the NCAR WRF model development group for providing the WRF model, the UCAR Developmental Tested Center (DTC) for the GSI DA system source code, and the Center for High-Performance Computing (CHPC) at the University of Utah for computer support. The NASA Award 80NSSC20K0900 is also acknowledged.

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
