# Peer review of "The impacts of assimilating Aeolus horizontal line-of-sight winds on numerical predictions of Hurricane Ida (2021) and a mesoscale convective system over the Atlantic Ocean"

_Atmospheric Measurement Techniques, 2022_

## Referee Comment (RC1)

Title: The impacts of assimilating Aeolus horizontal line-of-sight winds on numerical predictions of Hurricane Ida (2021) and a mesoscale convective system over the Atlantic Ocean

Author(s): Chengfeng Feng and Zhaoxia Pu

MS No.: amt-2022-341

This manuscript aims at investigating the influences of assimilating Aeolus horizontal line-of sight winds on forecasts of tropical cyclones and tropical convective systems. Better track predictions, intensity forecasts, precipitation structures are obtained in case studies. Overall, this manuscript is well organized and addressed its potential to improve numerical weather predictions. I suggest to accept the manuscript after some minor revisions.

1. The homogeneous isotropic horizontal ensemble localization scale is 110 km, and the vertical localization scale is 3 grid units. Any sensitive experiment supporting your configuration? Or add some references here.

2. Units are not uniformly written(e.g., 6-h ,48-hour, 700-hPa). Please Uniform the units throughout the manuscript and add a space between the value and unit. A dot is needed in the velocity unit(e.g., $m \cdot s^{-1}$). Revise it in the text and figures.

---

## Community Comment (CC4)

**Response to Referee #2's Comments: amt-2022-341**

*We appreciate the referee for his/her review and helpful comments. We have considered those comments very carefully and substantially revised our manuscript. A point-by-point response to the referee's comments is as follows.*

General comments

The manuscript describes the impact of Aeolus observations on the forecast of Hurricane Ida (2021) and a mesoscale convective system that occurred during NASA's CPEX-AW field campaign. The authors claim that the novelty of this investigation is that it evaluates the potential impact of the "near-real-time" assimilation of this data set.

As mentioned in the paper, previous work has focused on observing system experiments (OSEs). As a matter of fact, OSEs mimic near-real-time impacts as they use observations that were available at the time of the experiment. Thus, I don't understand what the limitations of earlier investigations are. In fact, these earlier studies used all the observations that were available for assimilation – not just conventional data.

*The previous observing system experiments did not examine how assimilating the Aeolus HLOS winds influences the dynamic and thermodynamic structures of the hurricane in detail, such as the vorticity and divergence in the inner core regions of the hurricane. Moreover, they did not examine the impacts on the mesoscale convective system. Our study attempted to fill these gaps.*

*In fact, the "conventional data" we referred to in the paper are all routinely assimilated data in the NCEP GDAS operational data assimilation system, including both conventional types of data and satellite data products. We have made clarifications in the text of the paper by mentioning them as "GDAS data".*

The findings of this study are limited to one single hurricane and one single mesoscale system, which makes it impossible to draw any general conclusions. Therefore, the evaluation of the impact of Aeolus is over-emphasized.

*Even if we have one main hurricane case in this paper, we did very comprehensive experiments. We performed the cycling data assimilations for hurricane Ida (2021), which can produce the forecasts initialized at different data assimilation cycles. We also carried out the sensitivity study of the initial time, horizontal, and vertical ensemble length scales for the experiments of hurricane Ida (2021). The results of horizontal and vertical were added to the revised version based on Referee #1's comments.*

*Meanwhile, we agree with you that more cases should be used in future studies. Therefore, we have mentioned these at the end of the revised paper to address your concern. Please see the sentences in the last paragraph of the revised manuscript.*

In addition, some of the technical elements are not well described. An example are the observations errors. Those are critical in data assimilation as reported in previous Aeolus data impact studies.

*We agree with you that the observation errors are critical for assimilating the Aeolus data. We considered the observational errors very carefully when performing our experiments. We did not use any default numbers for the observational errors. Instead, we used the error characteristics from the Aeolus data samples. We did assign different errors for Mie and Rayleigh winds (see updated Fig.2 and related text).*

The paper doesn't show any statistics of background/analysis departures either that are critical to evaluate the behavior of the assimilation algorithms used here.

*We added the statistics of background/analysis departures into the revised version of the paper (See Figure 3 and new Section 3.2).*

More detailed comments

L74: What is the meaning of near-real-time HLOS here – and what is different from the methodology used in previous studies (OSEs)? What is new in this study since only focuses on one single hurricane?

*"Near-real-time HLOS" refer to the type of Aeolus data we used. We have made the clarification in the text now.*

*As mentioned above, even if we have one main hurricane case in this paper, we did very comprehensive experiments. We performed the cycling data assimilations for hurricane Ida (2021), which can produce the forecasts initialized at different data assimilation cycles. We also carried out the sensitivity study of the initial time, horizontal, and vertical ensemble length scales for the experiments of hurricane Ida (2021).*

L99: Is the assimilated done in the parent domain, the inner domain or both?

*Data assimilation was performed in both domains. We clarified this in the revision.*

Fig.2: Are the Aeolus values observations or background simulations? Also, is the y-axis the instrument error or the observation error? It is not clear to this reviewer how the observation errors are estimated.

*The Aeolus values are observations. The y-axis is the instrument error provided by the Aeolus L2B product. We have revised this figure and the associated paragraph to explain how we estimated the observation errors.*

In a real-time environment, all available observations would be assimilated – not just conventional data. Why is it not done here? Are the reconnaissance data been assimilated?

*We agree. In fact, the "conventional data" we referred to in the paper include all conventional data available in the NCEP operational system, including both conventional types of data and satellite-derived data products. We have made clarifications in the text of the paper by mentioning them as "GDAS data". The reconnaissance data should be part of operational data; thus, they should be assimilated.*

[revised manuscript text omitted]

---

## Author Response (AR1)

**Response to Referees' Comments: amt-2022-341**

*We appreciate all the referees for their review and helpful comments. We have considered their comments very carefully and substantially revised our manuscript. A point-by-point response to referees' comments is as follows.*

**Referee #1**

General comments

This manuscript aims at investigating the influences of assimilating Aeolus horizontal line-of-sight winds on forecasts of tropical cyclones and tropical convective systems. Better track predictions, intensity forecasts, precipitation structures are obtained in case studies. Overall, this manuscript is well organized and addressed its potential to improve numerical weather predictions. I suggest accepting the manuscript after some minor revisions.

*We appreciate the reviewer for the helpful comments. We plan to address all your comments in the revision. Please read our point-by-point response and check the related part of the revised manuscript.*

Major comments

The homogeneous isotropic horizontal ensemble localization scale is 110 km, and the vertical localization scale is 3 grid units. Any sensitive experiment supporting your configuration? Or add some references here.

*Yes. Before finalizing the data assimilation configuration for experiments, we conducted sensitivity studies with various horizontal (55km, 110 km, and 220km) and vertical localization (3, 1.5, 6) scales. The sensitivity study results with horizontal and vertical localization scales have been summarized and added to the revised manuscript.*

*Please check Table 3, Figure 8, and the related paragraph in the track-changes version of the revised manuscript.*

Units are not uniformly written (e.g., 6-h ,48-hour, 700-hPa). Please uniform the units throughout the manuscript and add a space between the value and unit. A dot is needed in the velocity unit (e.g., $m \cdot s^{-1}$). Revise it in the text and figures.

*Thank you for pointing this out. Units in the manuscript have been checked and corrected. Please check the track-changes version of the revised manuscript.*

**Referee #2**

General comments

The manuscript describes the impact of Aeolus observations on the forecast of Hurricane Ida (2021) and a mesoscale convective system that occurred during NASA's CPEX-AW field campaign. The authors claim that the novelty of this investigation is that it evaluates the potential impact of the "near-real-time" assimilation of this data set.

*We appreciate the referee for his/her review and helpful comments. We have considered those comments very carefully and substantially revised our manuscript. A point-by-point response to the referee's comments is as follows.*

As mentioned in the paper, previous work has focused on observing system experiments (OSEs). As a matter of fact, OSEs mimic near-real-time impacts as they use observations that were available at the time of the experiment. Thus, I don't understand what the limitations of earlier investigations are. In fact, these earlier studies used all the observations that were available for assimilation – not just conventional data.

*The previous observing system experiments did not examine how assimilating the Aeolus HLOS winds influences the dynamic and thermodynamic structures of the hurricane in detail, such as the vorticity and divergence in the inner core regions of the hurricane. Moreover, they did not examine the impacts on the mesoscale convective system. Our study attempted to fill these gaps.*

*In fact, the "conventional data" we referred to in the paper are all routinely assimilated data in the NCEP GDAS operational data assimilation system, including both conventional types of data and satellite data products. We have made clarifications in the text of the paper by mentioning them as "GDAS data". Please check line 219, 261, 407, 430, and Table 3 in the track-changes version of the revised manuscript.*

The findings of this study are limited to one single hurricane and one single mesoscale system, which makes it impossible to draw any general conclusions. Therefore, the evaluation of the impact of Aeolus is over-emphasized.

*Even if we have one main hurricane case in this paper, we did very comprehensive experiments. We performed the cycling data assimilations for hurricane Ida (2021), which can produce the forecasts initialized at different data assimilation cycles. We also carried out the sensitivity study of the initial time, horizontal, and vertical ensemble length scales for the experiments of hurricane Ida (2021). The results of horizontal and vertical were added to the revised version based on Referee #1's comments. Please check Table 3, Figure 8, and the related paragraph in the track-changes version of the revised manuscript.*

*Meanwhile, we agree with you that more cases should be used in future studies. Therefore, we have mentioned these at the end of the revised paper to address your concern. Please see the sentences in the last paragraph of the revised manuscript.*

In addition, some of the technical elements are not well described. An example are the observations errors. Those are critical in data assimilation as reported in previous Aeolus data impact studies.

*We agree with you that the observation errors are critical for assimilating the Aeolus data. We considered the observational errors very carefully when performing our experiments. We did not use any default numbers for the observational errors. Instead, we used the error characteristics from the Aeolus data samples. We did assign different errors for Mie and Rayleigh winds (see updated Fig.2 and related text).*

The paper doesn't show any statistics of background/analysis departures either that are critical to evaluate the behavior of the assimilation algorithms used here.

*We added the statistics of background/analysis departures into the revised version of the paper (See Figure 3 and new Section 3.2).*

More detailed comments

L74: What is the meaning of near-real-time HLOS here – and what is different from the methodology used in previous studies (OSEs)? What is new in this study since only focuses on one single hurricane?

*"Near-real-time HLOS" refer to the type of Aeolus data we used. We have made the clarification in the text now.*
*As mentioned above, even if we have one main hurricane case in this paper, we did very comprehensive experiments. We performed the cycling data assimilations for hurricane Ida (2021), which can produce the forecasts initialized at different data assimilation cycles. We also carried out the sensitivity study of the initial time, horizontal, and vertical ensemble length scales for the experiments of hurricane Ida (2021).*
L99: Is the assimilated done in the parent domain, the inner domain or both?

*Data assimilation was performed in both domains. We clarified this in the revision. Please check line 214 and Figure 3 in the track-changes version of the revised manuscript.*

Fig.2: Are the Aeolus values observations or background simulations? Also, is the y-axis the instrument error or the observation error? It is not clear to this reviewer how the observation errors are estimated.

*The Aeolus values are observations. The y-axis is the instrument error provided by the Aeolus L2B product. We have revised this figure and the associated paragraph to explain how we estimate the observation error. Please check Figure 2 and associated paragraph in the track-changes version of our revised manuscript.*

In a real-time environment, all available observations would be assimilated – not just conventional data. Why is it not done here? Are the reconnaissance data been assimilated?

*We agree. In fact, the "conventional data" we referred to in the paper include all the conventional data available in the NCEP operational system, including both conventional types of data and satellite-derived data products. We have made clarifications in the text of the paper by mentioning them as "GDAS data". Please check line 219, 261, 407, 430, and Table 3 in the track-changes version of the revised manuscript. The reconnaissance data should be part of operational data; thus, they should be assimilated.*